# Reduction of Rapid Proliferating Tumour Cell Lines by Inhibition of the Specific Glycine Transporter GLYT1

**DOI:** 10.3390/biomedicines9121770

**Published:** 2021-11-25

**Authors:** Christine Garcia Bierhals, Alison Howard, Barry H. Hirst

**Affiliations:** Biosciences Institute, Faculty of Medical Sciences, Newcastle University, Newcastle upon Tyne NE2 4HH, UK; chrisbierhals@hotmail.com (C.G.B.); alison.howard@newcastle.ac.uk (A.H.)

**Keywords:** tumour, GLYT1, glycine transport, glycine metabolism, NSCLC, colorectal cancer, rapid proliferating tumour

## Abstract

Studies have highlighted the relevance of extracellular glycine and serine in supporting high growth rates of rapidly proliferating tumours. The present study analysed the role of the specific glycine transporter GLYT1 in supplying glycine to cancer cells and maintaining cell proliferation. GLYT1 knockdown in the rapidly proliferating tumour cell lines A549 and HT29 reduced the number of viable cells by approximately 30% and the replication rate presented a decrease of about 50% when compared to cells transfected with control siRNA. In contrast, when compared to control, GLYT1 siRNA had only a minimal effect on cell number of the slowly proliferating tumour cell line A498, reducing the number of viable cells by 7% and no significant difference was observed when analysing the replication rate between GLYT1 knockdown and control group. When utilising a specific GLYT1 inhibitor, ALX-5407, the doubling time of rapidly proliferating cells increased by about 8 h presenting a significant reduction in the number of viable cells after 96 h treatment when compared to untreated cells. Therefore, these results suggest that GLYT1 is required to maintain high proliferation rates in rapidly proliferating cancer cells and encourage further investigation of GLYT1 as a possible target in a novel therapeutic approach.

## 1. Introduction

Tumour cells present a different metabolism than normal cells utilising aerobic glycolysis to obtain energy even when oxygen levels are adequate, a metabolic mechanism known as the Warburg effect [1,2,3]. They also present an altered requirement for certain amino acids, different from normal cells [4]. Extracellular glycine and serine supplementation, for example, are essential for some tumours to maintain their rapid cell proliferation [4,5,6]. Glycine and serine starvation lead to a reduction in tumour size and increased survival in lymphoma and colorectal cancer mice models [7]. Rapidly proliferating tumour cells consume more glycine than slowly proliferating tumour cells or fast growing non-transformed cell lines and glycine was observed to be essential for maintenance of proliferation rates 4. Additionally, clinical studies have linked high glycine concentration in tumour tissues with poor patient prognosis in colorectal and breast cancers [8,9,10]. Taken together, this evidence suggests an important role for extracellular glycine in supporting rapid proliferation rate of some tumours.

Glycine is a small, versatile, and conditionally essential amino acid 1. It is known that extracellular glycine can also be utilised directly for purine and glutathione (GSH) synthesis therefore contributing to DNA replication and antioxidant reactions [4,11,12,13,14,15]. In vivo and in vitro experiments with radiolabelled glycine showed that extracellular glycine was incorporated directly into purine [14,15]. Rats with fibrocarcinoma and rats with mammary tumour infused with 13C-glycine presented a high concentration of radiolabelled GSH in tumour and also indicated that glycine was utilised in serine and cysteine synthesis [11,12,13]. GSH is the major cellular antioxidant and it has been related to cell proliferation and chemotherapy resistance [16,17]. The importance of glycine for tumour cell survival and proliferation increases when serine levels are limited. In vitro experiments demonstrated that fast growing tumour cells avidly consume serine from the medium [18]. Serine is then reduced to glycine in the serine biosynthesis pathway, a reaction catalysed by the mitochondrial enzyme serine hydroxymethyltransferase 2 (SHMT2) [18]. This reaction provides one-carbon units to form methyl-tetrahydrofolate (meTHF) supporting anaplerotic pathways [19]. Activity of SHMT2 has been suggested as the primary source of intracellular glycine [20]. Studies have also shown that, when serine levels are depleted, extracellular glycine can be utilised for serine synthesis in the reverse reaction catalysed by SHMT2 enzyme [18]. SHMT2 overexpression has been linked to poor outcome in cancer patients [4,21,22,23]. Additionally, the glycine cleavage system (GCS) has been related to cancer progression and can provide one-carbon units to support meTHF synthesis [15,24,25]. This way, extracellular glycine may contribute to rescue serine levels under nutrient deprivation, to anaplerotic pathways and also for DNA replication and antioxidant reactions possibly being crucial to support tumour cell proliferation.

Glycine can be transported into cells via the specific glycine transporters GLYT1 and GLYT2 (SLC6A9 and SLC6A5, respectively), and also by a variety of non-specific amino acid transporters: ATB^0,+^ (SLC6A14), PAT1 (SLC36A1), PAT2 (SLC36A1), SNAT2 (SLC38A2) and SNAT4 (SLC38A4) [26,27,28,29]. While GLYT2 is mainly expressed in neuronal tissues, GLYT1 is expressed in many peripheral organs [30,31]. In comparison to the latter, non-specific transporters that accept multiple amino acids as substrates, GLYT1 is highly specific for glycine and N-methylglycine derivatives such as sarcosine, demonstrating a high affinity transport [26]. GLYT1 is a Na^+^/Cl^-^ co-transporter that possesses 12 transmembrane domains [32]. Its expression is regulated by the activating transcription factor ATF4, itself linked to cancer proliferation and survival, being upregulated by amino acid starvation and during endoplasmic reticulum (ER) stress [33,34,35,36,37]. The specific glycine transporter GLYT1 is important in supplying glycine to meet cellular demands and maintaining intracellular levels of GSH during oxidative stress [12]. The Human Protein Atlas data shows that GLYT1 is overexpressed in some, but not all, tumour tissue samples from patients with melanoma, colorectal and lung cancer, among others [38,39,40,41,42]. Despite this information, little is known directly about GLYT1 relevance to tumour proliferation. To the best of our knowledge, this is the first publication regarding the relevance of GLYT1 transporter to tumour cell proliferation. The present study investigated the impact of GLYT1 downregulation on proliferation of rapidly and slowly proliferating tumour cell lines and also of a non-transformed cell line growing in minimal medium where levels of serine and other amino acids are low. The results showed that GLYT1 knockdown and also pharmacological inhibition of this transporter reduced glycine uptake and consequently cell proliferation of fast growing Non-Small Cell Lung Cancer (NSCLC) and colorectal tumour cells, A549 and HT29, respectively. In contrast, there was only a small or non-significant effect on a slow growing renal tumour cell, A498, and non-transformed human umbilical vein endothelial cells (HUVEC).

## 2. Materials and Methods

### 2.1. Cell Culture

A549 (ATCC^®^CCL-185TM), HT29 (ATCC^®^HTB-38TM), and A498 (ATCC^®^HTB-44TM) passage number between 23 and 53 cell lines were cultured in RPMI supplemented with 10% FCS, 2 mM L-glutamine, 100 U/mL penicillin, and 100 µg/mL streptomycin prior to treatments (Sigma-Aldrich, St. Louis, MO, USA). The HUVEC (Lonza, Basel, Switzerland) cell line, passage number between 3 and 6, was cultured in EBM™ Basal Medium (EGM) supplemented with Bovine Brain Extract, Ascorbic Acid, Hydrocortisone, Epidermal Growth Factor (hEGF), Foetal Bovine Serum (FBS), and Gentamicin/Amphotericin-B (Lonza). Cells were incubated at 37 °C, 5% CO_2_ in a sterilised, humidified incubator and grown on 75 cm^2^ canted neck cell culture flasks (Corning Inc., New York, NY, USA). The literature doubling times of these cell lines were experimentally confirmed and cells were classified as rapid or slow-proliferating cell lines.

For routine culture, medium was changed every 2–3 days. When confluent, monolayers were harvested using trypsin and passaged to new tissue culture 75 cm^2^ flasks. Cell suspensions were diluted and seeded at 1 × 10^6^ cells per flask for A549, HUVEC, and A498 cell lines and 3 × 10^6^ for HT29 cells. All tissue culture manipulation was undertaken in a class II laminar flow hood. For experimental purposes, cells were inoculated into multi-well plates. Cells were seeded into 96-well F-bottom culture plates (Greiner Bio One International GmbH, Stonehouse, UK) at final concentrations of 1 × 10^3^ cells/well for A549 and A498, and at 3 × 10^3^ cells/well for HT29 and HUVEC cells in a final volume of 100 µL of medium. When utilising 24 or 12-well plates all cells were seeded at concentrations of 3.5 × 10^4^ and 7 × 10^4^ cells/well in final volumes of 1 and 2 mL of medium, respectively.

### 2.2. Cell Treatments

#### 2.2.1. Knockdown Assays 

Cell lines were seeded onto 96-well dark culture plates or 12-well clear-coated culture plates in RPMI medium and incubated for 24 h. After this, the medium was replaced with antibiotic-free RPMI and the cells transfected with GLYT1 small interfering RNAs (siRNAs) si2991 or si2993 or negative control siRNA (Silencer^®^ Select Negative Control no2) from AmbionTM (Invitrogen, Thermo Fisher Scientific, Waltham, MA, USA). For transfection reactions, a mixture comprising 5 pmol of the appropriate siRNA, 4 µL Lipofectamine RNAi MAX, and 196 µL OptiMEM + GlutaMax (Gibco, Life Technologies, Carlsbad, CA, USA), previously incubated for 20 min at room temperature, was added to each well. Cell lines treated with Silencer^®^ Select Negative Control no2 (Life Technologies, Thermo Fisher Scientific, Waltham, MA, USA) were utilised as control. Cells were incubated at 37 °C for 72 h before RNA extraction and cell proliferation assays. The experiments were performed in triplicate.

#### 2.2.2. ALX-5407

The specific GLYT1 inhibitor ALX-5407, also known as N [3-(4’-fluorophenyl)-3-(4’-phenylphenoxy) propyl] sarcosine (NFPS) from Sigma-Aldrich, was utilised to inhibit GLYT1 function. ALX-5407 was diluted in DMSO to a stock concentration of 10 mM. DMSO is also a vehicle to facilitate the drug absorption into cells. A549, HT29, and A498 cells seeded in 24-well plates in RPMI supplemented with 5% FCS and Pen/Strep and HUVEC cells seeded in EGM were incubated for 24 h at 37 °C. Then, medium was changed, and cells were treated with 30, 60, 120, or 1000 nM ALX-5407 for 72 h to evaluate its effect on glycine uptake. To evaluate ALX-5407 effect on cell proliferation, cells were seeded in 96-well plates and treated with 120 nM ALX-5407 for 6, 24, 48, 72, 96, and 110 h. Control cells were incubated with normal medium containing the same concentration of DMSO (<0.01%).

#### 2.2.3. Tunicamycin 

Tunicamycin (Sigma-Aldrich) was diluted in DMSO to a stock concentration of 1mM. Cells were seeded in 96-well plates in RPMI for 24 h at 37 °C in an incubator, following 48 h incubation period with or without 120 nM ALX-5407. After this period the medium was changed to FCS-free medium and cells were treated for a further 12 h treatment with 0.1, 0.5, 1, 2, and 4 µM tunicamycin (TM) with or without 120 nM ALX-5407 to evaluate its effect on cell proliferation. Control cells were incubated with normal medium containing the same concentration of DMSO (<0.01%).

#### 2.2.4. Hypoxia

Cells were seeded in 96-well plates in RPMI and incubated for 24 h at 37 °C before hypoxia treatment. The medium was replaced with medium containing 30, 60, 120, or 1000 nM ALX-5407 and plates placed in a hypoxic chamber INVIVO2400 (Ruskinn Technology Ltd., Bridgend, UK), with 2% oxygen at 37 °C, for a further 72 h. Control cells were treated with the same final concentration of DMSO and exposure to hypoxia.

### 2.3. Cell Viability Assay

CellTiter Blue (CTB) assay (Promega UK, Southampton, UK) was used to verify cellular viability and proliferation. Briefly, cells were incubated with 100 µL RPMI medium and 20 µL of CTB at 37 °C for 3–4 h, and then fluorescence measured at 560 ex/590 em.

### 2.4. Cell Proliferation Assay

Cells were plated on coverslips and transfected with *GLYT1* specific or Negative siRNAs for 72 h. Click-it EdU assay (ThermoFisher, Waltham, MA, USA) was used to measure cell proliferation as described in the manufacturer’s protocol. Click-iT^®^ reaction cocktail was removed, and coverslips washed twice with 1 mL 3% BSA in PBS, then coverslips were lowered over slides containing a drop of Vectashield mounting medium with DAPI, to stain cell nuclei. Cell staining was then analysed on microscope Axio Imager with Apotome utilising filters appropriate for the Alexa Fluor^®^488 dye and DAPI.

### 2.5. Glutathione Measurement 

Following GLYT1 siRNA treatment in 96-well plates, the GSH-Glo™ Glutathione Assay (Promega UK Southampton, UK) was utilised for glutathione (GSH) measurement following the manufacturer’s protocol.

### 2.6. Real Time PCR (qPCR)

Total RNA was extracted from cells cultured in a 12-well plate following 72 h of transfection. The SV Total RNA extraction kit (Promega UK, Southampton UK) was utilised to extract total RNA following the manufacturer’s vacuum protocol. RNA yield, concentration, and purity were quantified on a spectrometer (Biomate-3) measuring absorbance at OD260/280 and 240. RNA integrity was determined using a Bioanalyzer (Supplier name), all RNAs had RIN > 8 and therefore were suitable for qPCR analysis. Total RNA (500 ng) was diluted with ultrapure water to a volume of 12.5 µL and 50 pmole of random hexamers (GE Bioscience) added. M-MLV Reverse Transcriptase RNase H+ (RT) reaction mix (Promega) was used according to the manufacturer’s protocol to obtain total cDNA.

The transcriptional profile of genes analysed was determined by quantitative real-time qPCR on a Lightcycler 480 (Roche Bioscience, Burgess Hill, UK). Sense and anti-sense primer pairs for target mRNAs were synthesised by IDT (Table 1). RT-products were diluted 1 in 5 in ultrapure water for qPCR reactions. Each qPCR mixture contained 2 µL diluted RT product, 1x Lightcycler 480 SYBR Green 1 Master Mix (Roche Bioscience, Burgess Hill, UK), and 0.5 µM forward and reverse primer. A hot-start cycling protocol (pre-incubation at 95 °C for 5 min) followed by 35–40 cycles of 5 s at 95 °C, 10 s at 58–62 °C, and 15 s at 72 °C, was followed by melt curve analysis over a 60–90 °C range to confirm specific product amplification. Each assay included duplicate sample reactions and negative control wells.

Serial dilutions (1:10) of a positive control, specific for each primer set, were included in all qPCR experiments to obtain a standard curve. The positives controls were obtained by cloning the PCR products in the cloning vector pGEM^®^-T Easy (Promega UK, Southampton UK). Vectors were transfected into *E. coli* DH5α and the plasmid DNA extracted and sequenced to confirm primer specificity. Plasmids were diluted to an optimal concentration and utilised to obtain standard curves, which were included in all experiments permitting the reaction efficiency calculation. Results with reaction efficiency of 85–105%, or 1.9–2.05, were considered suitable for the study. The qPCR data for each primer set were normalised to housekeeping genes Glyceraldehyde 3-phosphate dehydrogenase (*GAPDH*), DNA Topoisomerase I (*TOP1*), and ATP synthase F1 subunit beta (*ATP5B*) (PrimerDesign, Southampton, UK) using a normalisation factor generated from geometric averaging of the three reference genes utilising the GeNorm programme [43].

### 2.7. Glycine Uptake

Following cell line treatments, GLYT1kd or ALX-5407, glycine uptake experiments were performed. Medium was removed from cell culture plates, wells were washed three times with Krebs buffer, and incubated for 30–60 min at 37 °C. Cells were then incubated for 10 min at 37 °C with 5 mM glycine and 0.5 µCi/mL [2-3H]-glycine. Radioactivity in the samples was determined by liquid scintillation spectrophotometry using a Beckman LS5000 Liquid scintillation counter (Beckman-Coulter, Indianapolis, IN, USA). The amount of substrate was calculated using the following equation:

Amount of substrate = ((“ASTD”/“DPMSTD” “ × DPMSPL”))/“X”

Where, ASTD = amount of substrate in 100 µL of the standard, DPMSTD = average disintegration per minute of the radiolabelled substrate from 3 standards, DPMSPL = represents the disintegration per minute of a sample, Constant X = surface area per well, 3.8 and 1.9 for 12 and 24-well plates, respectively, used to express the result in cm^2^.

## 3. Results

### 3.1. GLYT1 Knockdown Reduces Glycine Uptake of Tumour Cells and Increases SHMT2 mRNA in Rapidly Proliferating Tumour Cell Lines

GLYT1 knockdown (kd) was first confirmed by qPCR. *GLYT1* mRNA expression was reduced by 86%, 79%, 93% when A549, HT29 and A498 cells, respectively, were treated with a specific GLYT1 siRNA, in comparison to control groups treated with Negative siRNA (Figure 1a). GLYT1kd cells demonstrated a significant reduction in glycine uptake from the extracellular medium when compared to Controls; glycine uptake was reduced by 35%, 31%, and 41% in A549, HT29 and A498 tumour cells, respectively (Figure 1b). These functional results suggest a reduction in GLYT1 protein levels, however, anti-GLYT1 antibodies were inefficient in Western blotting assays and immunolocalization studies, so an effect on protein abundance or localisation could not be confirmed. *SHMT2* mRNA was increased considerably, by 72% and 90%, respectively, in the rapidly proliferating cells, A549 and HT29, as a consequence of GLYT1kd when compared to controls (Figure 1c). In contrast, *SHMT2* mRNA abundance was unaltered by GLYT1kd in slowly proliferating A498 cells (Figure 1c). These results suggest that fast growing cancer cells are dependent on extracellular glycine supplied by GLYT1 and if this is compromised cells respond through upregulating SHMT2 in an attempt to enhance intracellular glycine synthesis.

### 3.2. GLYT1kd Reduces Proliferation and DNA Replication in Rapidly Proliferating Tumour Cells

*Ki67* mRNA expression, a proliferation marker [44,45], was reduced following GLYT1kd by 45% and 52% in rapidly proliferating A549 and HT29 cells, respectively (Figure 1d). In contrast, there was no effect on *Ki67* mRNA expression in slowly proliferating A498 cells with GLYTkd (Figure 1d).

The same pattern indicated by *Ki67* proliferation marker analysis was observed when cell proliferation assays were performed. GLYT1kd was associated with reduced numbers of rapidly proliferating A549 and HT29 cells, but with minimal effect on slowly proliferating A498 cells or rapidly proliferating non-transformed HUVEC cells (Figure 2a). These observations were quantified by CellTiter Blue (CTB) cell viability assay (Promega UK, Southampton, UK). Following GLYT1kd, there were significant reductions in cell viability of 35% and 31% for the rapidly proliferating cells, A549 and HT29, respectively, but only a minor or no effect on the number of viable A498 and HUVEC cells, respectively, by 72 h post-transfection, when compared to Controls (Figure 2b).

These data on the effects of GLYT1kd were further substantiated by quantifying incorporation of 5-ethynyl-2’-deoxyuridine (EdU), a nucleoside analogue of thymidine into DNA during active DNA synthesis (Click-It EdU assay, Thermo Fisher Scientific, Waltham, MA, USA) following GLYT1kd. When analysing the number of replicating nuclei, the proliferation index of rapidly proliferating cells A549 and HT29 following GLYT1kd indicated reductions of 47% and 68%, respectively, compared to control cells (Figure 3a). In contrast, no significant effect on the proliferation index of slowly proliferating A498 cells was observed following GLYT1kd (Figure 3a).

### 3.3. GLYT1kd Reduces GSH Levels of Slowly and Rapidly Proliferating Tumour Cell Lines

In contrast to the selective effect of GLYT1kd on proliferation of rapidly proliferating cells, GLYT1kd resulted in reductions in cellular GSH content of slowly and rapidly proliferating cell lines (Figure 3b). GSH content was reduced by 34%, 19%, and 15% in A549, HT29, and A498 cells, respectively, in response to GLYT1kd compared with Control cells (Figure 3b). These results suggest that the effect of GLYT1kd on cell proliferation may be explained by the impact of reduced glycine import on DNA replication in rapidly proliferating cell lines, rather than any further indirect effects on GSH homeostasis.

### 3.4. ALX-5407 Reduces Proliferation Rate of Rapidly Proliferating Cancer Cells

The specific GLYT1 inhibitor ALX-5407 was then investigated as a pharmacological approach to reducing the activity of GLYT1. Cells were treated with 30, 60, 120 nM and 1000 nM ALX5407, a specific GLYT1 inhibitor [46]. The concentration able to inhibit glycine uptake by approximately 50% was achieved with 120 nM ALX-5407 in the three tumour cells under investigation (Figure 4a). This concentration was then selected for further experiments. Cells were treated with 120 nM ALX-5407 for 110 h and the number of viable cells was assessed every 24 h. There was no significant difference observed between treatment and control groups in the slow growing renal tumour cell line A498 and the normal endothelial cell line HUVEC, whereas the fast-growing NSCLC and colorectal cell lines, A549 and HT29, demonstrated a reduction in cell proliferation after 96 h treatment with ALX-5407, of 29% and 23%, respectively (Figure 4b). At earlier time points, no significant reductions in cell proliferation were evident (Figure 4b). These observations suggest that during longer periods of incubation the GLYT1 transporter becomes essential to maintain proliferation rates of A549 and HT29 cells.

### 3.5. ALX-5407 Treatment Decreases Proliferation of Rapidly Proliferating Tumour Cell Lines during Hypoxia or ER Stress

Once a reduction in cell proliferation was only observed under prolonged incubation periods with ALX-5407 (Section 3.4), possibly due to nutrient starvation and/or increase in metabolic products, further experiments to assess different stress types were conducted. Tumour cells were subject to hypoxia, by incubation in a 2% oxygen chamber, or ER stress, induced by tunicamycin treatment, in association with 120nM ALX-5407 treatment for 72 h. ALX-5407 treatment did not influence proliferation of A498 cells under these stress conditions in comparison to controls (vehicle only) (Figure 5). However, with ER stress, ALX-5407 treatment significantly increased the deleterious effect of tunicamycin on HT29 cell proliferation when compared to tunicamycin treatment alone, causing a further reduction of 23% in the number of viable cells (Figure 5a). Hypoxia reduced cell proliferation of A459, HT29, and A498 cell lines by 39.3%, 30.9%, and 32.6%, respectively, when compared to normoxia condition. Treatment with ALX-5407 during hypoxia further inhibited the proliferation of A549 cells in comparison to hypoxia alone (Figure 5b). ALX-5407 under normal levels of oxygen resulted in no change in proliferation rates in any cell line. These results suggest that inhibition of GLYT1 may increase the susceptibility of tumours to different stress conditions depending on the cell type.

## 4. Discussion

Rapidly proliferating tumour cells require extracellular glycine, which is used in purine and GSH synthesis, to support their proliferation rate [4]. Glycine can also be utilised, under serine limitation, as substrate of the enzyme SHMT2 to support serine synthesis, a crucial amino acid for tumour cell proliferation [13,18]. Additionally, increased glycine concentration in tumour tissue of patients has been linked to poor patient prognosis [10]. Glycine can be transported into cells via a specific glycine transporter GLYT1 [26,30]. Previous studies in this group showed that under stress conditions extracellular glycine transported via GLYT1 supports GSH levels and cell survival [12]. Glycine also acts as a neurotransmitter and GLYT1 inhibitors have been tested for treatment of central nervous system disorders, such as schizophrenia [47]. Therefore, corroborating with the preliminary hypothesis, the data from the present work showed that inhibition of glycine transport via GLYT1 presented an impact on tumour cell proliferation. This novel observation indicates the relevance of GLYT1 for proliferation of colorectal and NSCLC rapidly proliferating tumour cells.

GLYT1 knockdown efficiently reduced *GLYT1* mRNA expression and significantly decreased glycine uptake in A549, HT29, and A498 tumour cell lines when compared to control cells. As a consequence, there was a decrease in cell proliferation for the rapidly proliferating NSCLC and colorectal tumour cell lines, A549 and HT29, respectively. In contrast, there was only a minor effect on the slowly proliferating renal cancer cell line, A498. Moreover, GLYT1kd did not affect cell proliferation of the rapidly proliferating non-transformed cell line, HUVEC. These data suggest that GLYT1 transporter is crucial to support tumour rapidly proliferating rates. These results are in accordance with previous work from Jain et al. [4] who observed that extracellular glycine is essential to maintain cellular proliferation of rapidly proliferating tumour cells, but not to support slowly proliferating cancer cells and rapidly proliferating non-tumour cell proliferation rates when these cells were growing in the minimal medium, RPMI. Additionally, both A549 and HT29 cell lines showed upregulation of SHMT2 mRNA, a key enzyme in intracellular glycine generation, following knockdown of GLYT1. Possibly this was an attempt to increase glycine synthesis to meet cellular demands. However, the upregulation of SHMT2 was insufficient to rescue cell proliferation. Similar results were observed by Jain et al. [4] where glycine starvation reduced cell proliferation of rapidly proliferating tumour cells even with increased SHMT2 expression [4]. This way, the results indicate that extracellular glycine transported into cells via GLYT1 transporter is essential to support cellular proliferation of rapidly proliferating tumour cells as NSCLC and colorectal tumour cells.

As previously discussed, extracellular glycine can be utilised in purine and GSH synthesis. Radiolabelled glycine uptake experiments indicate that extracellular glycine provides C2N directly into purine synthesis to glycinamide ribonucleotide (GAR) synthesis [4,15,48]. Glycine can be incorporated directly into GSH and also via one-carbon units [4,13,14]. Interestingly, a study utilising rat models for breast cancer observed that extracellular glycine was utilised in serine synthesis, which was then utilised in the trans-sulphuration pathway to produce cysteine that was incorporated into GSH in the tumour tissue [13]. GSH is a major cellular antioxidant and in some tumours is related to cell survival during stress conditions, such as during chemotherapy treatment, and therefore conferring resistance to these drugs [16]. GSH is also important for cell cycle progression and supports DNA replication by maintaining reduced glutaredoxin levels, which is essential for ribonucleotide reductase activity and then for deoxyribonucleotide synthesis [17]. Glycine catabolism occurs via GCS producing meTHF that in some tumour cells is fully oxidised generating NADPH, which also contributes to REDOX reactions [48]. GCS activity has also been related to tumour growth and/or proliferation and it increases pyrimidine levels [23,24]. Purine and pyrimidine nucleotide bases are the building blocks of DNA and RNA. This way, as expected, GLYT1kd and consequent reduction in glycine uptake influenced GSH synthesis and DNA replication, which may explain GLYT1 influence in cell proliferation. Rapidly proliferating tumour cells with GLYT1kd presented a major reduction in DNA replication in comparison to control cells. GLYT1kd also reduced GSH levels of all tumour cells. This result is in agreement with a previous study, which showed that Caco-2 colorectal cells require extracellular glycine transported by GLYT1 to support GSH levels and cell survival under oxidative stress [12]. Therefore, a major role for extracellular glycine supplied via GLYT1 in purine synthesis, DNA replication, and in antioxidant reactions might be taken into consideration.

When tumour cells were treated with a specific GLYT1 inhibitor, ALX-5407, an effect on cell proliferation was only observed during stress conditions. A significant reduction in cell proliferation of A549 and HT29 cells was detected following longer periods of incubation (>92 h) which may lead to nutrient depletion and/or an increase of metabolic products in the medium, considering the increase in cell numbers and that the medium was only replenished every 48 h. Under oxidative and ER stress conditions, A549 and HT29 cell lines treated with ALX-5407 for 72 h demonstrated a significant reduction in cell proliferation in comparison to cells not exposed to the inhibitor. It is known that, although low, the cationic lipid-based reagent Lipofectamine RNAiMAX presents a significant cytotoxicity to cells during transfection with siRNA [49] which was also observed in the present work experiments. Moreover, the cells in this study were incubated with the transfection reagent for 72 h, the longer period suggested by the manufacturer, in a medium without serum supplementation, to assure a downregulation in protein level, which may also increase the stress conditions to the cells during the GLYT1 knockdown experiments. All together, these data suggest that extracellular glycine supplied by GLYT1 only becomes relevant for rapidly proliferating tumour cells to maintain their proliferation rates during stress. It is known that some tumours provide a stressful environment due to low vascularisation, therefore facing hypoxia and nutrient starvation, or due to chemotherapy treatment [16,50].

In addition to purine and GSH synthesis, glycine is also involved in angiogenesis signalling. Under normal plasma glycine concentrations, VEGF induces GLYT1 expression and glycine uptake that seems to be involved in an mTOR signalling pathway which promotes angiogenesis [51]. Interestingly, studies have shown that increased concentrations of glycine, about 10 times the normal plasma levels (~250 µM), inhibit angiogenesis and subsequent tumour growth, counteracting the effect of VEGF [52,53,54]. A clinical study observed that colorectal cancer patients with increased glycine concentration in tumour tissue presented poor prognosis [10]. Moreover, lung cancer patients present a decreased concentration of glycine in plasma and an elevated concentration in tumour tissue in comparison to normal individuals and non-transformed tissue [55]. This way, GLYT1 is possibly involved in reducing glycine levels in plasma and increasing glycine levels in tumour tissue that would influence angiogenesis. Thus, GLYT1 inhibition may contribute to an increase in sensitivity of rapidly proliferating tumours to stress, reduction in DNA replication, and possibly would contribute to a reduction in angiogenesis.

Therefore, the present study suggests that GLYT1 might be considered as a promising new target for cancer treatment of rapid proliferative tumours and encourages further in vivo studies. Since extracellular glycine influences GSH synthesis, GLYT1 may also be considered as a target to overcome chemotherapy resistance. Additionally, the availability of specific GLYT1 inhibitors already in use in human could present exciting opportunities.

## Figures and Tables

**Figure 1 biomedicines-09-01770-f001:**
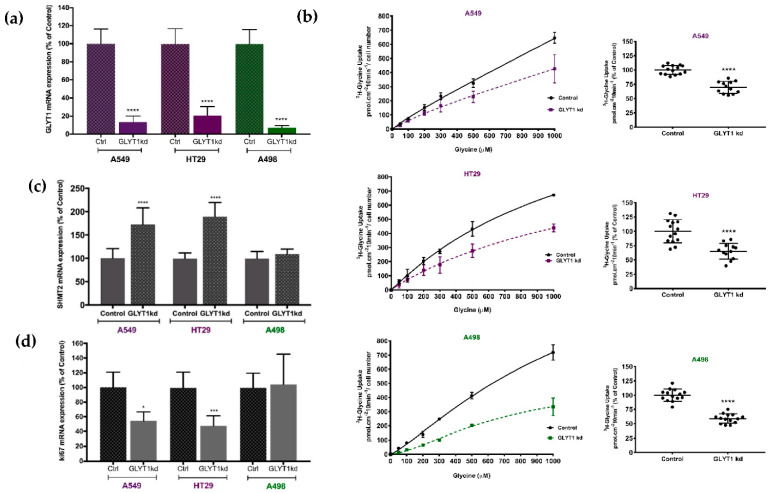
GLYT1 knockdown efficiency and effect on proliferation marker ki67. (**a**) Following GLYT1kd there was a significant reduction of *GLYT1* mRNA abundance for all cells analysed when compared to control groups. (**b**) GLYT1 downregulation significantly reduced total glycine uptake measured by [2-3H]-glycine uptake analysis for all tumour cell lines in this study. (**c**) qPCR analysis showing upregulation of *SHMT2* mRNA expression following GLYT1kd in comparison to control. (**d**) qPCR analysis of the influence of GLYT1kd on the proliferation marker ki67 mRNA expression in comparison to control. Statistical analysis comparing control and GLYT1kd was performed by unpaired t-test considering significance at *p* < 0.05 (*), *p* < 0.001 (***), and *p* < 0.0001 (****).

**Figure 2 biomedicines-09-01770-f002:**
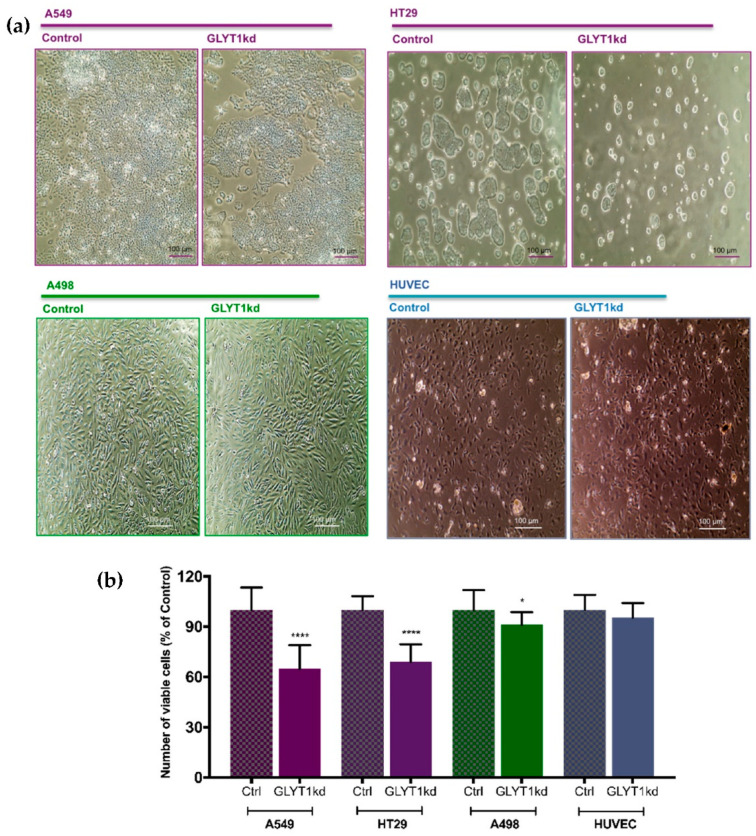
GLYT1 knockdown effect on cell proliferation. (**a**) 20× microscopy images of A549, HT29, A498, and HUVEC cell lines representative of controls and GLYT1kds. (**b**) Analysis of cell proliferation following 72 h transfection with Negative siRNA (control) or *GLYT1* siRNA (GLYT1kd) utilising CTB assay. Data presented as percentage of control as mean ± SEM, *N* = 3 and *n* = 15 per group. Statistical analysis comparing control and GLYT1kd was performed by unpaired *t*-test considering significance at *p* < 0.05 (*) and *p* < 0.0001 (****).

**Figure 3 biomedicines-09-01770-f003:**
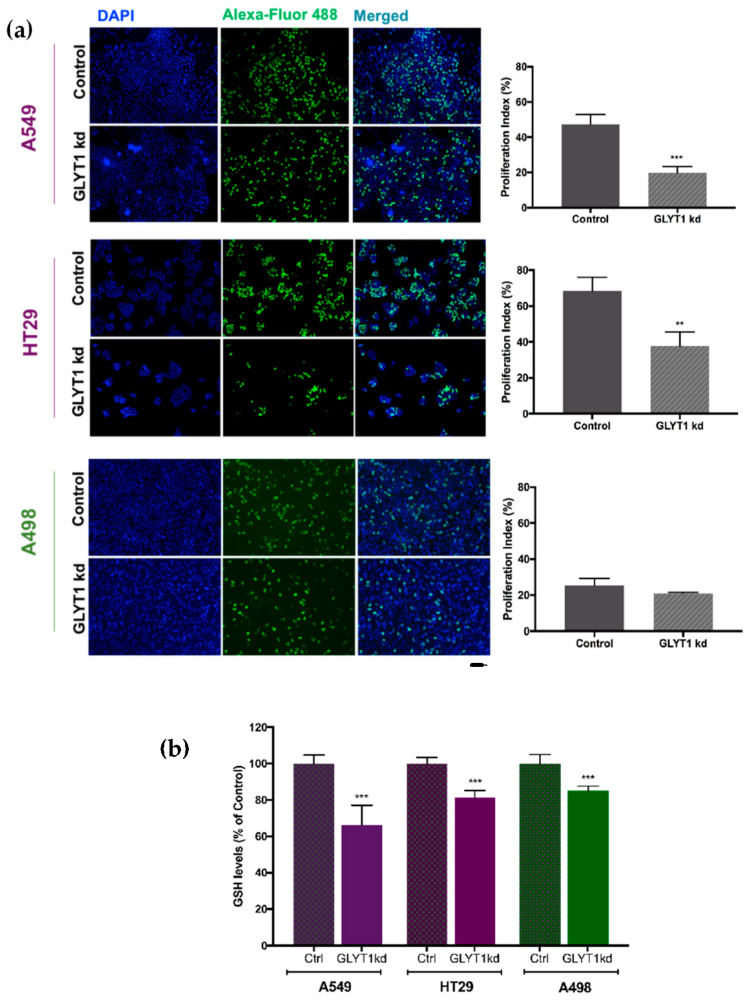
GLYT1 knockdown effect on DNA replication and on GSH levels. (**a** Left panel) Fluorescence microscopy images of total nuclei stained with DAPI (blue) and proliferating nuclei stained with EdU labelled with AlexaFluor ^®^488 (green) utilising Click-It EdU assay. Bar indicates 50 µm. (**b** Right panel) Four photographs were randomly selected for each group, Control and GLYT1kd, and number of total nuclei and proliferating nuclei (EdU) were counted utilising ImageJ program. The percentage of proliferation index was calculated (proliferating nuclei/total nuclei number ×100). Data represent mean ± SEM, *n* = 4. Statistical analysis comparing treatment (GLYT1kd) and control of each cell line was performed by unpaired t-test where *p* < 0.01 (**) and *p* < 0.001 (***). (**b**) GSH levels analysed following 72 h transfection with Negative siRNA (control) or GLYT1 siRNA (GLYT1kd) by GSHGlo assay (Promega). Data presented as percentage of control mean ±SEM, *N* = 2, *n* = 6 per group. Statistical analysis comparing treatment (GLYT1kd) and control of each cell line was performed by unpaired t-test where *p* < 0.001 (***).

**Figure 4 biomedicines-09-01770-f004:**
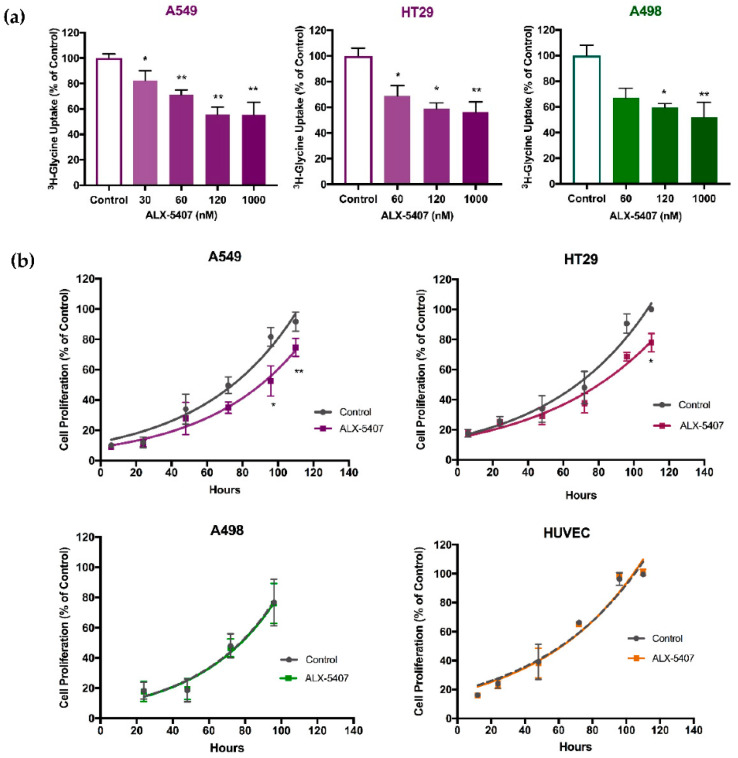
ALX-5407 treatment reduced total glycine uptake in all cells analysed and proliferation of A549 and HT29 cells. (**a**) Effect of ALX-5407 treatment on total glycine uptake as percentage of control. Cells were incubated with 5 µM glycine in association with [2-3H]-glycine 0.5 µCi/mL in Krebs buffer for 10 min. Data presented as mean ± SEM, *N* = 3 and *n* = 9 per group. Statistical analyses were performed by one-way ANOVA with Dunnett’s post hoc analysis considering significance at *p* < 0.05 (*) and *p* < 0.001 (**). (**b**) Growth curve of A549, HT29, A498, and HUVEC cell lines treated with 120nM ALX-5407 over 110h (coloured lines) in comparison to untreated cells (grey lines). Data are presented as mean ± SEM, *N* = 3 and *n* = 9. Statistical analysis comparing treated and untreated cells for each time point was performed by unpaired t-test considering *p* < 0.05 (*) and *p* < 0.001 (**).

**Figure 5 biomedicines-09-01770-f005:**
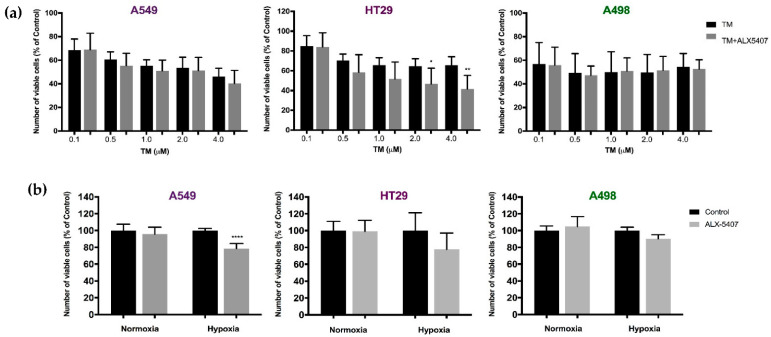
ALX-5407 effect on cell proliferation under stress conditions. (**a**) Number of viable cells counted following 72 h incubation with or without 120 nM ALX-5407 and further 12h treatment with 0.1, 0.5, 1, 2, and 4 µM tunicamycin (TM) alone or in association with 120 nM ALX-5407. Data are presented as ± SEM, *N* = 3 and *n* = 12. Statistical analyses were performed by one-way ANOVA with Tukey’s post hoc analysis where *p* < 0.05 (*) and *p* < 0.01 (**). (**b**) Number of viable cells counted following 72 h treatment with 120nM ALX-5407 incubated under normoxic and hypoxic conditions as percentage of untreated (control) cells according to CTB assay analysis. Data are presented as ± SEM, *N* = 3 and *n* = 9. Statistical analyses were performed by unpaired t-test where *p* < 0.0001 (****).

**Table 1 biomedicines-09-01770-t001:** Primers utilised in end-point PCRs and qPCRs. Description includes gene name, NCBI identification number, sequence, exon position, annealing temperature, and product size. Numbers in superscript at the start and end of each primer sequence indicate the base location in the named cDNA.

Gene Name	NCBI id	Primer Sequence (with Base Numbering)	Exon Position	Annealing Temp. (°C)	Product Size (bp)
Glyt1 ^1^	NM_001024845 (transcript variant 3)	Sense: ^580^CCATGTTCAAAGGAGTGGGCTA^601^	4:5	60	76
		Antisense ^2^: cggc^651^TGACCACATTGTAGTAGATGCCG^629^	5		
SHMT2 ^2^	NM_005412 (transcript variant 1)	Sense: ^1043^CAAGACTCTTGCAGGGGCCAG ^1063^	7	60	129
		Antisense: ^1171^GATGGGAACACGGCAAAGTTG ^1151^	8		
Ki67 ^3^	NM_002417 (transcript variant 1)	Sense: ^2375^TACATGTGCCTGCTCGACCC^2394^	12	60	125
		Antisense: ^2499^CTGCGGTTGCTCCTTCACTG^2480^	13		

^1^ This primer pair will detect all *GLYT1* transcript variants except the long non-coding variant 6. Primer locations in GLYT1a (variant 3) only are given. ^2^ This primer pair will detect all *SHMT2* transcript variants. Primer locations in variant 1 only are given. ^3^ This primer pair will detect Ki67 transcript variants 1 and 2. Variants 1 and 2 both encode the same protein. Primer locations in variant 1 only are given.

## Data Availability

Not applicable.

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
