# Peer review of "Reduction of Rapid Proliferating Tumour Cell Lines by Inhibition of the Specific Glycine Transporter GLYT1"

_biomedicines, 2021, doi:10.3390/biomedicines9121770_

Round 1

Reviewer 1 Report

This is a well written paper with the results being clearly presented. The topic is of interest to a broad audience. However, there are several weaknesses that must be addressed:

  1. Given that the siRNA efficiency cannot be measured by protein levels, the authors use qPCR to validate knockdown. Have they used multiple qPCR primers located at different sites of the gene so as to ensure that they are truly getting gene silencing as compared to alternate isoforms?

  1. Was the mRNA abundance of GLYT1 in all four cell lines about the same (highly proliferating versus slower cells)?

  1. Why do the authors use qPCR to show SHMT2 upregulation? Can they not use antibodies to show this by westerns?

  1. Does GLYT1 kd cause any other effects rather than just SHMT2 upregulation? Does it also cause upregulation of the other transporters?

  1. Figure 2b: maybe the authors would be able to see an effect upon stress induction (withdrawal of glycine/glycine starvation)

  1. Figure 3B: The authors claim, based on figure 3b, that since GLYT1kd causes decreased GSH levels in all three cell lines, but affects replication only in the two highly proliferative cell lines, that it is a replication-specific effect rather than a GSH homeostasis ‘indirect’ effect. I am not sure I agree, especially given that the decrease in GSH level seems much less pronounced in the slower cell line as compared to A549 and HT29.

  1. Figure 5B: Why does the inhibitor not cause a decrease in normoxia, even in the highly proliferative cells? This is not consistent with the siRNA effect- and argues against the model that GLYT1 is required for tumor cell proliferation in the absence of stress

  1. Lastly, given that most of these findings were previously known, I am not really sure what these experiments add to the field. I would suggest that the authors include a paragraph addressing this very specifically, either in the introduction or conclusion.

Reviewer 2 Report

The manuscript entitled “Reduction of rapid proliferating tumour cell lines by inhibition of the specific glycine transporter GLYT1” by Garcia Bierhals et al. presents data regarding the role of the glycine transporter GLYT1 in rapidly- and slowly- proliferating cancer cell lines. Their results present strong evidence that the knockdown of GLYT1 impairs glycine uptake in cancer cell lines, leading to a decreased proliferation that is very strong in rapidly-proliferating cancer cell lines (A549 ad HT29). The manuscript is well written and well presented, but would benefit from one additional experiment (major point) and further clarifications (minor points).

Major point

  1. In Figure 4, the authors show a decreased proliferation of A549 and HT29 cells upon treatment with ALX-5407. This difference seems to disappear in Figure 5B in the normoxia condition, but is significant in hypoxia. Is it because of the timepoint chosen (72h)? Would it be more convincing to also do this experiment at a later timepoint (96-120 hours) like in Figure 4B?

Minor points

  1. In Figure 3A, I would add the “Edu” in the Alexa488 column to make is easier for the reader, even if it is indicated in the figure legend.
  2. In Section 3.5 of the results, I would briefly remind the readers why you are looking at the role of hypoxia and ER stress.
  3. The authors say on line 322: “Hypoxia reduced cell proliferation of all cell lines tested presenting a higher effect on A549 cells when compared to normoxia condition (data not shown).” I think these results would benefit from being included in the paper.
  4. In Figure 5 legend, a clarification might be needed and a typo corrected (ALX-5404 should be ALX-5407?): “(a) Number of viable cells counted following 12h treatment with 0.1, 0.5, 1, 2 and 4µM tunicamycin (TM) alone or in association with 72h 120nM ALX-5404 as percentage  of untreated cells according to CTB analysis”. Is the treatment with 12h with TM, followed by 72h with ALX-5407? Are the drugs put together at the same time and there is a typo in 12h?

Round 2

Reviewer 1 Report

The authors have addressed most of my concerns.

Author Response

Corrected to ALX-5407.  Our apologies for introducing this new error.
